# Automatic Spray Trajectory Optimization on Bézier Surface

**Wei Chen [1,2], Junjie Liu [1], Yang Tang [3,\*] and Huilin Ge [1]**

[1]  School of Electronics and Information, Jiangsu University of Science and Technology, Zhenjiang 212003, China; cwchenwei@aliyun.com (W.C.); junjiel_mtr@163.com (J.L.); gehuilin404@163.com (H.G.)

[2]  School of Automation, Southeast University, Nanjing 210096, China

[3]  School of Science, Jiangsu University, Zhenjiang 212013, China

\*  Correspondence: ty800117@ujs.edu.cn

**Abstract:** The trajectory optimization of automatic spraying robot is still a challenging problem, which is very important in the whole spraying work. Spray trajectory optimization consists of two parts: spray space path and end-effector moving speed. A large number of spraying experiments have proved that it is very important to find the best initial trajectory of spraying. This paper presents an automatic spray trajectory optimization that is based on the Bézier surface. Spray the workpiece for Bezier triangular surface modeling and find the best initial trajectory of the spraying robot, establish the appropriate spraying model, plan the appropriate space path, and finally plan the trajectory optimization along the specified painting path. The validity and practicability of the method presented in this paper are proved by an example. This method can also be extended to other applications.

**Keywords:** initial trajectory; trajectory optimization; Bézier surface

## 1. Introduction

With the development of intelligence, multi-intelligence systems have received extensive attention and application. It can be seen that intelligent automation is widely used in painting robots [1,2]. Surface modeling of the sprayed workpiece is the first step in the trajectory optimization of the spray painting robot and the key to designing the spray path of the spray painting robot. At present, there are two main types of surface modeling methods of sprayed workpiece for off-line programming system of spray painting robot: (1) Surface modeling method based on CAD (Computer Aided Design) model. The modeling method based on the CAD model is that the CAD model data of the workpiece have been obtained before the surface modeling, and the spraying path of the spray painting robot can be planned according to the CAD model of the workpiece. (2) Surface modeling method based on the workpiece scanning system. If there is no CAD model data for a workpiece, or if the surface shape of actual workpiece does not match the CAD model data, then the workpiece needs to be scanned to obtain its new CAD data. The surface of the workpiece is approximated by a simple plane, sphere, cylinder, or other parametric surface, allowing spray path planning on these parametric surfaces.

In the previous work, we focused on the trajectory optimization of complex curved surface. In the reference [3], the Surface modeling method based on the CAD model is adopted. The CAD model data of the workpiece is obtained, and can plan the spraying path of the spraying robot according to the CAD model of the workpiece. The trajectory optimization of spray painting robot for complex curved surface based on the exponential mean Bézier method is proposed. The advantage is that it does not need to split the complex curved surface.

During the off-line programming operation for spray painting robot, after the surface modeling for the workpiece is finished, the following work is to optimize the trajectory of spray painting robot [4,5]. Since the point on the trajectory is a six-dimensional vector in the Cartesian coordinate system, it is very complicated to express it in mathematical expressions and the solving process is very difficult. Therefore, the general idea of trajectory optimization for spray painting robot is usually as follows: First, find the spatial path of the spray painting robot on the workpiece surface, and then find out the optimal time sequence along the specified spatial path. That is, the consistency of the paint thickness on the workpiece surface is the highest and the spray painting time is the shortest at what speed the end effector sprays along the specified spatial path. According to this idea, the optimized trajectory for spray painting consists of two parts: the spatial path of the spray painting operation and the moving speed of the end effector.

On the other hand, a large number of spray painting experiments have shown that the uniformity of the paint thickness can be significantly improved. That is, the spray painting effect can be improved by optimizing the initial trajectory of the spray painting robot in the initial stage of spray painting [6–8]. In other words, finding the best initial trajectory for spray painting is critical to the further trajectory optimization. Therefore, the trajectory optimization for spray painting robot can be divided into the following four steps: (1) Finding the optimal initial trajectory of spray painting robots. (2) According to the geometric features of sprayed workpiece surface, establish a suitable spray painting model. (3) Plan an appropriate spatial path for spray painting. (4) Plan trajectory optimization along the specified painting path.

The initial trajectory selection, the establishment of spraying model and path planning are all the bases of trajectory optimization. This research is based on the assumption that the workpiece CAD model was not acquired in advance. The innovation is that the Bézier triangular surface modeling method is adopted under the CAD model data without the workpiece. Firstly, the Bézier surface is analyzed and the method for searching the optimal initial trajectory of the spray painting robot is given according to the features of the Bézier triangular surface. Subsequently, the spray painting model of Bézier surface is established, and the mathematic expression of paint thickness at a certain point on the Bézier surface is given. Finally, the optimized trajectory of Bézier surface is obtained by using the ideal point method in the mathematical programming with the uniformity of paint thickness and the shortest spray painting time as optimization objectives along the specified spray painting path. The advantage of this method is that a good initial path of automatic spraying is determined at the beginning of the spraying process, which can significantly provide uniformity of coating thickness, that is, improve the spraying effect.

## 2. Bézier Triangular Surface Modeling Method of Sprayed Workpiece

Since the Bernstein polynomial has many superior properties, it is widely used in parametric polynomial curve surfaces of many forms. Based on the features of the sprayed workpiece surface, the Bézier triangular surface is constructed by using the Bernstein polynomial as the basis function.

**Definition 1.** *There is an arbitrary given triangle on the plane whose vertices $T_1$ $T_2$, $T_3$ are in the counterclockwise direction. Point P is any point in the plane where the triangle $T_1T_2T_3$ is located.*

Subsequently, we define:

$$u_1 = \frac{[PT_2T_3]}{[T_1T_2T_3]}, u_2 = \frac{[T_1PT_3]}{[T_1T_2T_3]}, u_3 = \frac{[T_1T_2P]}{[T_1T_2T_3]} \tag{1}$$

In the Equation (1), $[T_1T_2T_3]$ represents the directed area of the triangle $T_1T_2T_3$; When $T_1$, $T_2$, $T_3$ is counterclockwise, $[T_1T_2T_3]$ represents the area of the triangle $T_1T_2T_3$, that is $[T_1T_2T_3] = S$; When $T_1$, $T_2$, $T_3$ is clockwise, $[T_1T_2T_3]$ represents the opposite number of the area of the triangle, that

is, $[T_1 T_2 T_3] = -S$. Afterwards, we call $(u_1, u_2, u_3)$ as the area coordinate of Point $P$, recorded as $P = (u_1, u_2, u_3)$. Triangle $T_1 T_2 T_3$ is also called as coordinate triangle, as shown in Figure 1.

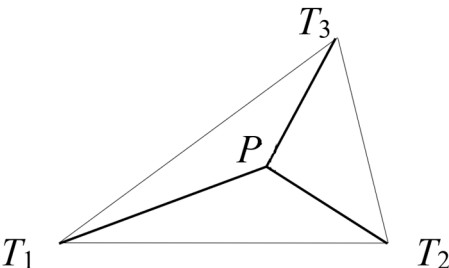

**Figure 1.** The Area Coordinate of Point $P$ on the Coordinate Triangle.

**Definition 2.** *Suppose the area coordinate of point $P$ on the coordinate triangle $T$ is $(u_1, u_2, u_3)$, and then we define:*

$$B_{i,j,k}^n(P) = \frac{n!}{i!j!k!} u_1^i u_2^j u_3^k , \quad i + j + k = n \tag{2}$$

As the Bernstein basis function $((n+1)(n+2)/2$ in all), They have the following properties:

(1) Non-negative: $B_{i,j,k}^n(P) \geq 0, P \in T, i + j + k = n$;

(2) Normative: $\sum\limits_{i+j+k=n} B_{i,j,k}^n(P) = 1$;

where, according to the triangular theorem we can have:

$$(a + b + c)^n = \sum\limits_{i+j+k=n} \frac{n!}{i!j!k!} a^i b^j c^k, \quad a, b, c \in R, \quad n \in N \tag{3}$$

For any $P$, $u_1 + u_2 + u_3 = 1$. Let $a = u_1, b = u_2, c = u_3$, we can have $\sum\limits_{i+j+k=n} B_{i,j,k}^n(P) = 1$.

Use the any straight line parallel to one side of the triangle to equate the remaining two sides of the coordinate triangle $T$ into $n$ segments, then the three parallel lines will divide the triangle into $n^2$ small congruent triangles, thus we can make the $n$-time subdivision of the coordinate triangle $T$, recorded as $S_n(T)$. Subsequently, we call each small triangle is the sub-triangle of $S_n(T)$. The vertices of the sub-triangle $((n+1)(n+2)/2$ in all) are called as the node that subdivides $S_n(T)$. The coordinates of the sub-nodes are as follows:

$$\left(\frac{i}{n}, \frac{j}{n}, \frac{k}{n}\right), i + j + k = n \tag{4}$$

Abbreviated as:

$$P_{i,j,k} = \left(\frac{i}{n}, \frac{j}{n}, \frac{k}{n}\right) \tag{5}$$

**Definition 3.** *Suppose $b_{i,j,k}(i + j + k = n)$ is any real number, we call*

$$B^n(P) = B^n(u_1, u_2, u_3) = \sum\limits_{i+j+k=n} b_{i,j,k} B_{i,j,k}^n(P) \tag{6}$$

As the $n$-time Bézier facet on the coordinate triangle $T$, $b_{i,j,k}$ $(i + j + k = n)$ as the Bernstein coefficient of the Bézier triangular surface, $P_{i,j,k} = (P_{i,j,k}; b_{i,j,k})$, $i + j + k = n$ as the control vertice of the Bézier triangular surface. We call the patch linear continuous function, which is linear on the sub-triangle of $S_n(T)$ and is the value $b_{i,j,k}$ at node $P_{i,j,k}$ as the control grid of the Bézier triangular surface.

In particular, for any function $f : T \to R$, the Bernstein coefficient is taken as:

$$b_{i,j,k} = f(\frac{i}{n}, \frac{j}{n}, \frac{k}{n}) \tag{7}$$

Then we call

$$B^n(P) = B^n(u_1, u_2, u_3) = \sum_{i+j+k=n} f(\frac{i}{n}, \frac{j}{n}, \frac{k}{n}) B_{i,j,k}^n(P) \tag{8}$$

as the n-time Bernstein triangular polynomial of $f$ on $T$.

Where, in order to simplify the derivation process, three shift operators $E_1$, $E_2$, $E_3$ are introduced, which are defined as:

$$E_1 b_{i,j,k} = b_{i+1,j,k} \tag{9}$$

$$E_2 b_{i,j,k} = b_{i,j+1,k} \tag{10}$$

$$E_3 b_{i,j,k} = b_{i,j,k+1} \tag{11}$$

Subsequently, $E_1^i E_2^j E_3^k b_{0,0,0} = b_{i,j,k}$, and we have:

$$B^n(P) = \sum_{i+j+k=n} \frac{n!}{i!j!k!} u_1^i u_2^j u_3^k (E_1^i E_2^j E_3^k b_{0,0,0}) \tag{12}$$

With the trinomial expansion, Equation (12) can be expressed as:

$$B^n(P) = (u_1 E_1 + u_2 E_2 + u_3 E_3)^n b_{0,0,0} \tag{13}$$

Accordingly, we have:

$$B^n(T_1) = E_1^n b_{0,0,0} = b_{n,0,0} \tag{14}$$

$$B^n(T_2) = E_2^n b_{0,0,0} = b_{0,n,0} \tag{15}$$

$$B^n(T_3) = E_3^n b_{0,0,0} = b_{0,0,n} \tag{16}$$

Here, we call point $P_{n,0,0} = (1, 0, 0; b_{n,0,0})$, $P_{0,n,0} = (0, 1, 0; b_{0,n,0})$, $P_{0,0,n} = (0, 0, 1; b_{0,0,n})$ the corner points of the triangular surface.

When $u_1 = 0$, $u_3 = 1 - u_2$. Substituting into Equation (6), then we have:

$$B_{i,j,k}^n(P) = \frac{n!}{j!(n-j)!} u_2^j (1 - u_2)^{n-j} = B_j^n(u_2) \tag{17}$$

Subsequently,

$$B^n(0, u_2, 1 - u_2) = \sum_{j=0}^n b_{0,j,n-j} B_j^n(u_2), 0 \le u_2 \le 1 \tag{18}$$

The boundary of a triangular surface is the n-time Bézier curve with the boundary of triangular surface control grid as the control polygons.

According to the definition of Bézier triangular surface modeling, when $n = 2$, the quadratic Bézier triangular surface generated by six control vertices is:

$$\begin{aligned} B^2(P) &= \sum_{i+j+k=2} b_{i,j,k} \frac{2!}{i!j!k!} u_1^i u_2^j u_3^k \\ &= u_1^2 b_{200} + u_2^2 b_{020} + u_3^2 b_{002} + 2u_1 u_2 b_{110} + 2u_1 u_3 b_{101} + 2u_2 u_3 b_{011} \end{aligned} \tag{19}$$

The above Equation (19) can be further expressed as a quadratic form:

$$B^2(P) = \begin{pmatrix} u_1 & u_2 & u_3 \end{pmatrix} \begin{pmatrix} b_{200} & b_{110} & b_{101} \\ b_{110} & b_{020} & b_{011} \\ b_{101} & b_{011} & b_{002} \end{pmatrix} \begin{pmatrix} u_1 \\ u_2 \\ u_3 \end{pmatrix} \tag{20}$$

The resulting quadratic Bézier triangular surface and its control network projection are shown in Figure 2.

When $n = 3$, the cubic Bézier triangular surface that was generated by ten control vertices is:

$$\begin{aligned} B^3(P) &= \sum_{i+j+k=3} b_{i,j,k} \frac{3!}{i!j!k!} u_1^i u_2^j u_3^k \\ &= u_1^3 b_{300} + u_2^3 b_{030} + u_3^3 b_{003} + 3u_1^2 u_2 b_{210} + 3u_1 u_2^2 b_{120} + 3u_1^2 u_3 b_{201} \\ &\quad + 3u_1 u_3^2 b_{102} + 3u_2 u_3^2 b_{012} + 3u_2^2 u_3 b_{021} + 6u_1 u_2 u_3 b_{111} \end{aligned} \tag{21}$$

The resulting cubic Bézier triangular surface and its control network projection are shown in Figure 3.

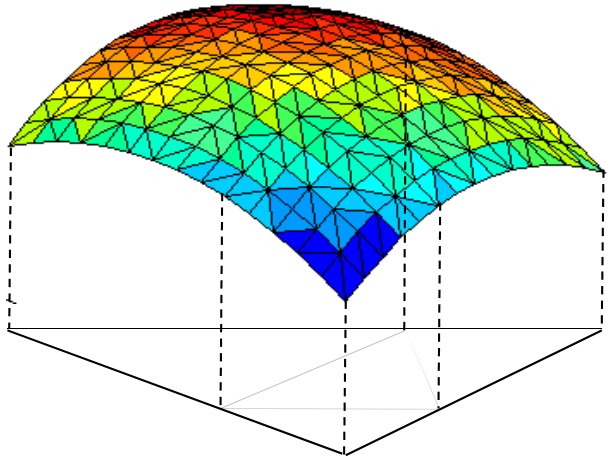

**Figure 2.** Quadratic Bézier Triangular Surface and Its Control Network Projection.

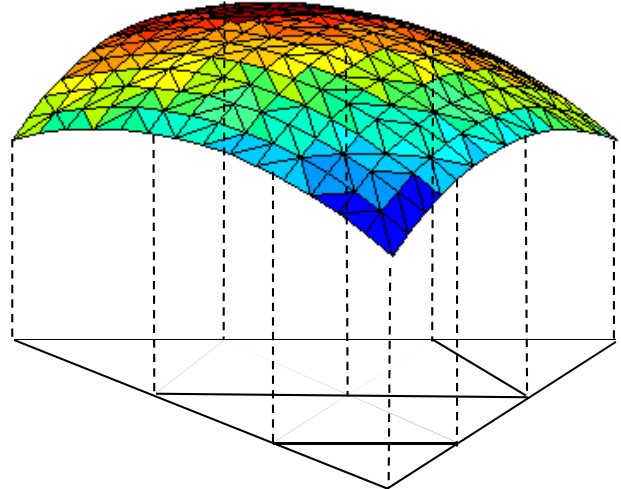

**Figure 3.** Cubic Bézier triangular surface and its control network projection.

## 3. Optimal Initial Trajectory Selection for Automatic Spraying on Bézier Surface

### 3.1. Bézier Surface Definition and Correlative Properties

In particular, the Bézier surface mainly includes a tensor product surface on a rectangular domain and a triangular surface on a triangular domain.

$m \times n$-time Bézier surface can be represented as:

$$B(u,v) = \sum_{i=0}^{m} \sum_{j=0}^{n} B_{i,m}(u) B_{j,n}(v) V_{i,j} \tag{22}$$

where, $B_{i,m}(u), B_{j,n}(v)$ are the u-direction m-time and v-direction n-time Bernstein basis function. $V_{i,j}(i = 0,1,\cdots,m; j = 0,1,\cdots,n)$ is the control vertice or Bézier point of the curved surface. The control vertices form $m+1$ and $n+1$ control polygons along the v-direction and u-direction, respectively, which together form a curved control grid or a Bézier grid.

The properties of the Bézier surface are as follows:

(1)  The four corner points of the Bézier grid are the four corner points of the Bézier surface:

$$B(0,0) = V_{0,0}, B(1,0) = V_{m,0}, B(0,1) = V_{0,n}, B(1,1) = V_{m,n} \tag{23}$$

(2)  The outermost vertex of Bézier grid defines the four borders of the Bézier surface and it has the following characteristics at the boundary, as shown in Table 1:

**Table 1.** Characteristics at the Boundary of Bézier Grids.

|  | (0,0) | (1,0) | (0,1) | (1,1) |
|---|---|---|---|---|
| $B$ | $V_{0,0}$ | $V_{m,0}$ | $V_{0,n}$ | $V_{m,n}$ |
| $\frac{\partial B}{\partial u}$ | $m\Delta^{1,0}V_{0,0}$ | $m\Delta^{1,0}V_{n-1,0}$ | $m\Delta^{1,0}V_{0,n}$ | $m\Delta^{1,0}V_{m-1,n}$ |
| $\frac{\partial B}{\partial v}$ | $n\Delta^{0,1}V_{0,0}$ | $n\Delta^{0,1}V_{m,0}$ | $n\Delta^{0,1}V_{0,n-1}$ | $n\Delta^{0,1}V_{m,n-1}$ |

(3)  Affine invariance: The Bézier surface is not changed under affine transformation.
(4)  'Symmetry': The control vertices in opposite order define the same Bézier surface.
(5)  Convex hull: The Bézier surface is always located in the three-dimensional convex hull generated by its control vertex.
(6)  Move vertice $V_{i,j}$, it will have the largest effect on the point $B(i/m, j/n)$, corresponding to $u = i/m, v = j/n$.

### 3.2. Optimal Initial Trajectory Selection

A large number of spray painting practical applications show that in the beginning of the spray painting operation if we can determine a good initial trajectory of spray painting robot, the uniformity of paint thickness can be significantly improved. That is to say, the spray painting effect can be improved. It can also reduce the spray painting time, improve the spray painting efficiency, and reduce the rate of paint waste at the same time. At present, the optimal initial trajectory selection method of the existing painting robot is to use the plane cutting method to take the obtained cross line as the initial trajectory of the painting robot [9–11]. This method can improve the spray painting effect to a certain extent, but the randomness is large and the spray painting time cannot be optimized. In this paper, the initial trajectory selection method that is based on geodesic curvature can not only improve the spray painting effect (paint thickness uniformity), but also can improve the spray painting efficiency (reduce spray painting time).

The curvature of a certain point on the painting path can be divided into two kinds: The first one is used to characterize the bending degree of the painting path passing through the point along normal

vector of the surface, which is called the normal vector curvature. The second is used to characterize the extent to which the spray trajectory bends to the boundary line of the surface, called geodesic curvature. As shown in Figure 4, Figure 4a is a zero-geometric curvature trajectory. That is, the paint thickness is uniform on the both side of the painting path. Figure 4b is the spray painting trajectory whose geodesic curvature is a constant. It is obvious that more paint is accumulated in the direction where the trajectory is bumped. Figure 4c shows the painting path of the variable geodesic curvature. The paint thickness on both sides of the trajectory is not uniform. It can be seen that the changing rate of the geodesic curvature at each point on the painting path has an influence on the paint thickness. In order to improve the consistency of the paint thickness, the changes in geodesic curvature at each point on the trajectory must be taken into account when selecting the initial spray painting trajectory.

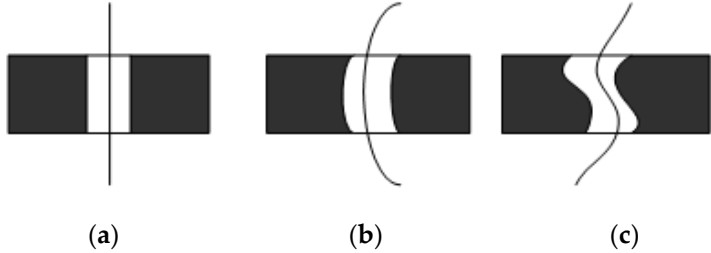

(**a**) (**b**) (**c**)

**Figure 4.** The Influence of Geodesic Curvature on the Consistency of Paint Thickness.

From the example shown in Figure 4, it can be seen that the zero-geometric curvature curve should be selected when determining the initial trajectory, and the initial trajectory also determines the geodesic curvature of the subsequent trajectory. The process of selecting the initial trajectory for spray painting can be divided into two steps: 1) Determining the relative position of the initial spray painting trajectory and the boundary of the workpiece surface. 2) Selecting the direction of the initial spray painting trajectory. As shown in Figure 5a, the initial trajectory is a geodesic, but the geometric curvature is very high when the offset curve passes the area near the apex of the cube. In Figure 5b, the initial trajectory is also the geodesic, but the surface is symmetrically bisected into two parts with the same Gaussian curvature being integral. Thus, in determining the position of the initial trajectory, it is necessary to select a position that minimizes the geodesic curvature of the offset curve.

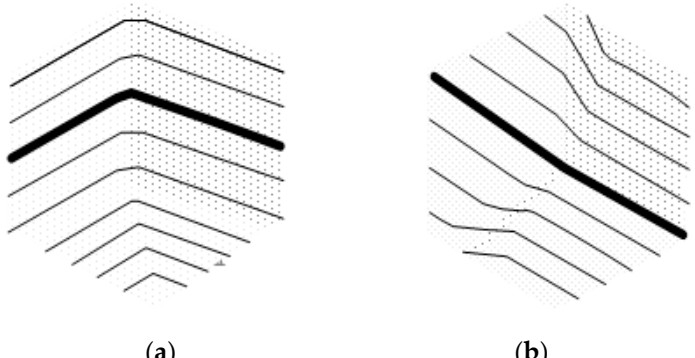

(**a**) (**b**)

**Figure 5.** The Influence of the Relative Position of Geodesic Curve and Surface Boundary on Subsequent Trajectory.

### 3.2.1. Determining the Relative Position of Initial Trajectory

As shown in Figure 6, let a segment on the smooth initial trajectory $\alpha_0$ as $C_{st}$. We use $\alpha_0(t_0)$ and $\alpha_0(t_1)$ to represent the two endpoints of $C_{st}$. The offset curve of line segment $C_{st}$ is generated by measuring the distance between the offset curve and the initial trajectory according to the geodesic lines $\gamma_{t_0}$ and $\gamma_{t_1}$, which are perpendicular to the initial trajectory at points $\alpha_0(t_0)$ and $\alpha_0(t_1)$. Assume that $C_{of}$ is the offset curve with a distance $\Delta$ of $C_{st}$ and the offset distance $\Delta$ is less than the focal length

of $\alpha_0$. Here, we only need to consider the integration of the geodesic curvature along $C_{of}$. Accordingly, we can assume that the surface is continuous. Subsequently, we can assume that $C_{of}$, $\gamma_{t_0}$, and $\gamma_{t_1}$ are all smooth curves. Assume that $\phi$ is a region enclosed by boundaries $C_{st}$, $C_{of}$, $\gamma_{t_0}$, and $\gamma_{t_1}$, and an arbitrary smooth curve connecting $\gamma_{t_0}(\Delta)$ and $\gamma_{t_1}(0)$ is $C_{dia}$. Suppose that the region bounded by $C_{st}$, $\gamma_{t_0}$, and $C_{dia}$ is $\phi_1$, its boundary $\partial\phi_1$ contains the curves $C_{st}$, $\gamma_{t_0}$, $C_{dia}$, and their corresponding directions. Similarly, $\phi_2$ denotes the region bounded by $C_{of}$, $\gamma_{t_1}$, and $C_{dia}$, and $\partial\phi_2$ is the boundary of $\phi_2$.

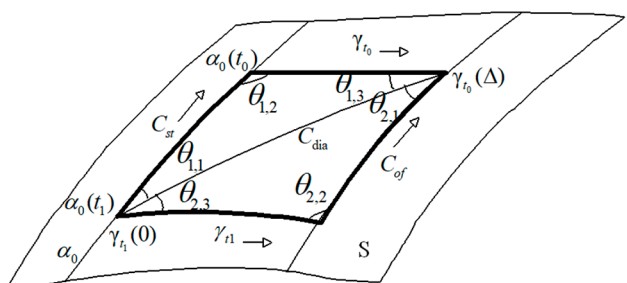

**Figure 6.** The Relationship between the Curvature Integral of Geodesic Surface and the Gaussian Curvature of Surfaces.

By applying the Gauss-Bonnet formula to the triangular regions $\phi_1$ and $\phi_2$, we can obtain that:

$$\iint_{\phi_i} K d\sigma + \oint_{\partial\phi_i} k_g ds = \sum_{j=1}^{3} \theta_{i,j} - \pi, \quad i = 1, 2 \tag{24}$$

In the expression above, $K$ is the Gaussian curvature of the surface $\phi_i$. $k_g$ is the geometric curvature of the triangle boundary $\partial\phi_i$. $\theta_{i,j}$ is the $j$-th interior angle of the boundary $\partial\phi_i$. As $\gamma_{t_0}$ and $\gamma_{t_1}$ are geodesic, so the integrals $\int_{\gamma_{t_0}} k_g ds$ and $\int_{\gamma_{t_1}} k_g ds$ are zero. Subsequently, we have:

$$\oint_{\partial\phi_1} k_g ds + \oint_{\partial\phi_2} k_g ds =$$
$$\left( \int_{C_{st}} k_g ds - \int_{\gamma_{t_0}} k_g ds - \int_{C_{dia}} k_g ds \right) + \left( \int_{\gamma_{t_1}} k_g ds + \int_{C_{dia}} k_g ds - \int_{C_{of}} k_g ds \right) \tag{25}$$

$$\oint_{\partial\phi_1} k_g ds + \oint_{\partial\phi_2} k_g ds = \int_{C_{st}} k_g ds - \int_{C_{of}} k_g ds \tag{26}$$

Obviously, $\iint_{\phi_1} K d\sigma + \iint_{\phi_2} K d\sigma = \iint_{\phi} K d\sigma$, $\gamma_{t_0}$, and $\gamma_{t_1}$ are perpendicular to the seed curve, so we have: $\theta_{1,1} + \theta_{2,3} = \theta_{1,2} = \frac{\pi}{2}$, Substituting into expression (26), then we sum the triangular regions $\phi_1$ and $\phi_2$:

$$\int_{C_{of}} k_g ds = \iint_{\phi} K d\sigma + \int_{C_{st}} k_g ds \tag{27}$$

Finally, if the initial trajectory is a geodesic, then:

$$\int_{C_{of}} k_g ds = \iint_{\phi} K d\sigma \tag{28}$$

### 3.2.2. Selecting the Direction of Initial Path

In order to ensure that the time along the initial trajectory is the least, when selecting the spatial direction of the initial trajectory, it is necessary to select a curve from numerous geodesic curvature and Gaussian curvature curve on the surface whose integrals are equal to that of an initial trajectory. We can give the definition of surface height first, and then determine the direction of the initial trajectory according to the definition. The surface height is the sum of the longest geodetic curve (straight

geodesic) perpendicular to the initial trajectory that extends from the initial trajectory to both sides, as shown in Figure 7. The optimal initial trajectory is the initial trajectory corresponding to the minimum surface height of the surface. With the same spray painting distance, the reciprocating spray painting time along the optimal initial trajectory is the least (Figure 8). What is more, the paint consistency is the best and the paint consumption is also the least.

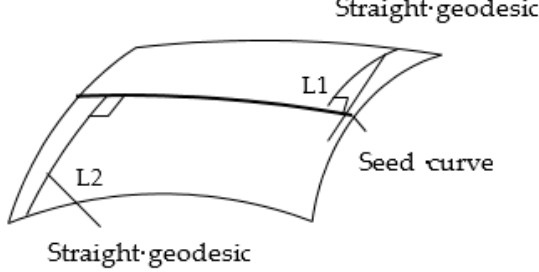

**Figure 7.** Surface Height Measurement Method.

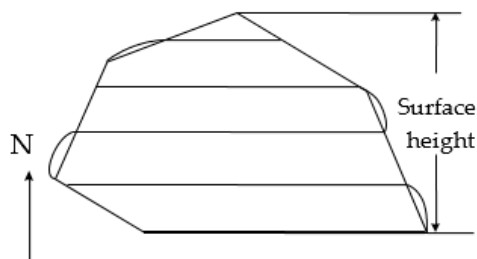

**Figure 8.** Relationship between Surface Height and Reciprocating Spray Time.

## 4. Automatic Spray Space Path Generation on Bézier Surfaces

Under normal circumstances, using the Bézier method to generate the surface is more complex. It is more complex if we perform optimization of spraying trajectory directly on Bézier surface. On the other hand, during the spraying operation, the distance between the end effector and the workpiece surface is always constant and perpendicular to the workpiece surface. In this case, the end effector of the spray equipment is essentially reciprocating on the isometric surface of the workpiece surface. Therefore, according to this idea, we can first find its isometric surface according to the shape of the surface and then perform optimization of spraying trajectory on the isometric surface. It should be noted that, strictly speaking, it is the discrete point array of the isometric surface of the Bézier surface but not the isometric surface of Bézier surface that we will find out according to this method. In the spatial path planning of the spray painting robot, we only need to find the discrete points on the path essentially and then fit out the spray path using the corresponding mathematical methods according to the accuracy requirements.

A U-direction or V-direction Bézier curve on the Bézier surface boundary is used as a benchmark and it is discretized under a certain precision. Subsequently, specify a constant painting distance h and find the equidistant point of the discrete points on the Bézier surface in the direction of the normal vectors of the discrete points along the curve. After connecting these equidistant points with a smooth curve, we can find an equidistant line of a Bézier curve on the boundary line of the Bézier surface. By the same token, the Bézier curves with the same distance are specified on the Bézier surface (which is the distance between two adjacent painting paths). In the same way, the same method can be used to find the discrete point array of the equidistant surfaces of the Bézier surface. Afterwards, we use the cubic Cardinal spline curves to connect each discrete point array. The adjacent two segments of cubic Cardinal spline curve segments are connected by a Hermite spline curve, so that the specified painting path can be obtained.

## 5. Trajectory Optimization on the Bézier Surface

In the actual off-line programming process of spray painting robot, the following factors should be taken into consideration when performing the trajectory optimization for spray painting robot on curved surface: (1) Mathematical model of surfaces. (2) Spray painting model on curved surface. (3) The expression of paint thickness at a point on the surface. (4) Mathematical expression of optimized trajectory on surface and its solution. In essence, the trajectory optimization for spray painting robot is actually a multi-objective optimization problem with constraints. There are many constraints in this problem, such as the error of paint thickness, the path of the end effector, the moving speed, the surface shape of the sprayed workpiece, parameters of the end effector, air pressure, paint viscosity, and so on. Accordingly, how to deal with the constraint function effectively in order to guide the algorithm searching is the key of trajectory optimization problem [12–15]. On the other hand, there are many optimization objectives, such as minimum spray painting time, smallest variance of paint thickness, minimum paint consumption, the highest paint utilization rate, the least inflexion of the robot path, and so on. In these spray painting optimization objectives, the objective function of the trajectory optimization is not independent of each other. They are often coupled with each other and in a state of competition. As a result, it is very difficult to obtain the exact solution of the multi-objective trajectory optimization problem of spray painting robot.

In order to obtain high-efficient painting path, the ideal method is to make certain assumptions. That is, in the case that the error is allowed, a number of parameters are assumed to remain unchanged in the process of spray painting. Only the main factors in the spray painting process are taken into account. Such kind of idea makes the trajectory optimization of spray painting robot greatly simplified and it also makes the multi-objective optimization of spray painting trajectory with constraints being easy to be solved.

When solving the optimization problem of spray painting trajectory on Bézier surfaces, we will simplify and solve the problem according to the ideal above. The specific idea is as follows: After the Bézier triangular surface modeling method is used to obtain the parametric surface, a simple paint deposition rate model is established. On this basis, a general spray painting model on the Bézier surface is derived and the mathematical expression of paint thickness at an arbitrary point is also derived. Finally, the optimal spray painting speed and spray painting time are selected as optimization objectives. After the multi-objective optimization function of the spray painting robot on Bézier surface is derived, the appropriate mathematical programming method is used to obtain the solution and the optimized trajectory of spray painting robot on Bézier surface can also be obtained.

The spatial distribution model of coating, the cumulative rate of coating function diagram and free surface trajectory optimization method have been described in the previous work [3,16]. After spraying a curved surface $S$, assuming that the average thickness of the surface is $q_d$, the coating thickness at any point $s$ is $q_s$, the deviation of the maximum coating thickness is $q_w$, and then we have:

$$\max_{s \in S}(|q_d - q_s|) \leq q_w \tag{29}$$

Assuming that the maximum coating thickness is $q_{\max}$ and the minimum coating thickness is $q_{\min}$, the maximum deviation angle between the normal vector of all sampling points and the normal vector of the surface is $\beta_{th}$, then the coating thickness at any point s can satisfy the following inequality:

$$q_{\min} \cdot \cos \beta_{th} \leq q_s \leq q_{\max} \tag{30}$$

The coating thickness at any point s satisfies the requirement (29), then we have:

$$|q_s - q_d| \leq q_w, s \in S \tag{31}$$

then:

$$q_{\max} - q_d \leq q_w \tag{32}$$

Further:

$$q_d - q_{\min} \cdot \cos \beta_{th} \leq q_w \tag{33}$$

If Equation (32) can be satisfied, then the maximum deviation angle $\beta_{th}$ can be calculated with Equation (33). That is, for any surface, if the deviation angle $\beta$ satisfies $\beta \leq \beta_{th}$, then the coating thickness at any point on the curved surface can satisfy Equation (29).

## 6. Experimental Part

### 6.1. Experimental Verification

The sprayed workpiece is shown in Figure 9. According to the topology of the spray workpiece, the workpiece is divided into three parts for processing, which are basin bottom, basin side, and basin edge, respectively. In the three parts, the basin bottom and the basin edge are all flat, and the surface of these parts can be directly generated by the control vertice. The basin side needs to be divided into two patches for processing, which are both arc. Where 10 control vertices are taken at different positions on each patch, and the cubic Bézier triangular surfaces are generated using the algorithm in Part 2. The Bézier triangular surface generation software system is written in VC ++ language. The curved surface modeling diagram of the workpiece is shown in Figure 10.

After the sprayed workpiece is modeled by the Bézier triangular surface technique, the U-direction spatial path and the V-direction spatial path of the workpiece surface are obtained according to the method for generating the spatial path of the spray painting robot on the Bézier surface that is presented in this paper. U-direction Spatial Path and V-direction Spatial Path as shown in Figures 11 and 12.

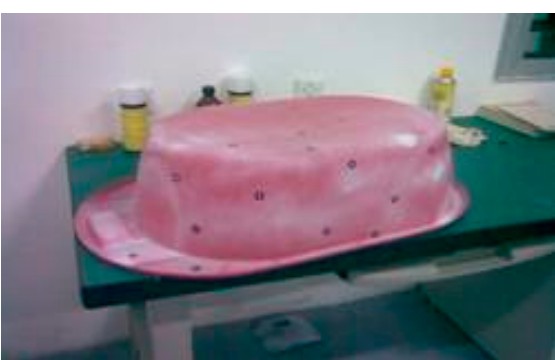

**Figure 9.** Sprayed Workpiece.

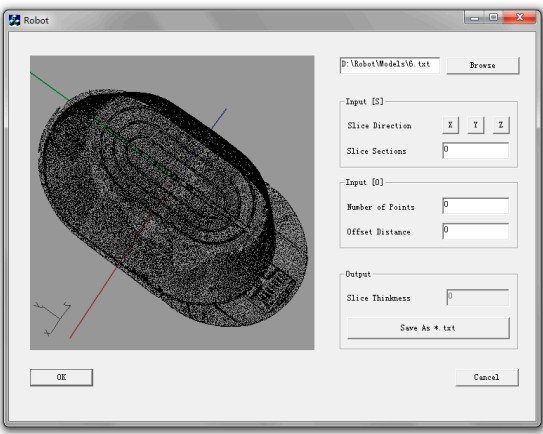

**Figure 10.** The Curved Surface Modeling Diagram of the Workpiece.

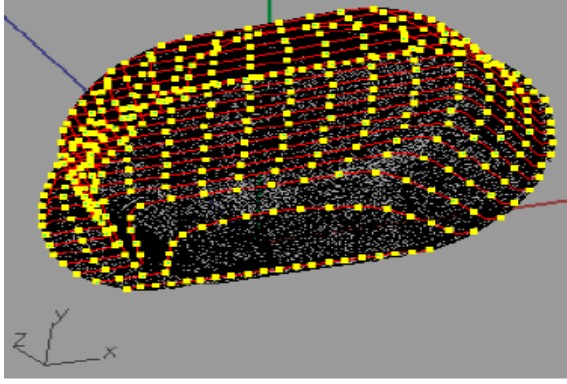

**Figure 11.** U-direction Spatial Path.

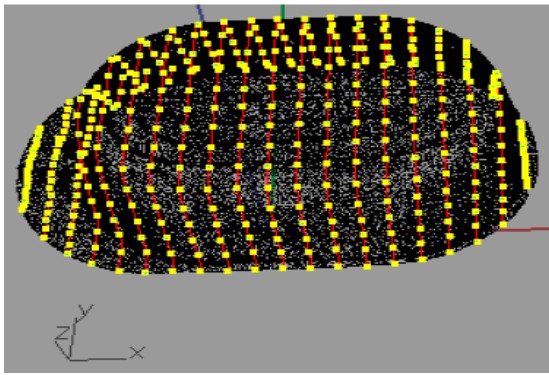

**Figure 12.** V-direction Spatial Path.

Assuming that the ideal paint thickness is $qd$ = 50 μm, the error of the maximum paint thickness is $qw$ = 10 μm, the bottom radius of the conical paint sprayed by the end effector R = 60 mm. According to the Spatial distribution model of coating, after performing the spray painting experiment on the plate, the paint deposition rate obtained by the experimental data is:

$$f(r) = \frac{1}{15}(R^2 - r^2) \ \mu m/s \tag{34}$$

After obtaining the optimized trajectory on the plane, the spray painting rate (at uniform speed) of the spray painting robot can be obtained as: V = 256 mm/s.

After obtaining the spatial painting path, trajectory optimization is carried out along the specified painting path according to the trajectory optimization method for spray painting robot on the Bézier surface in Section 5. At the same time, according to the initial trajectory selection method for spray painting robot on the Bézier surface in Section 3, the initial trajectory is selected. There are 432 discrete points in the discrete point array in U-direction, and the path between every two discrete points is divided into 10 segments. There are 402 discrete points in the discrete point array in the V-direction, and the path between every two discrete points is divided into 10 segments. The parameters of the algorithm are as follows: the ideal paint thickness $qd$ = 50 μm, maximum allowable error $qw$ = 10 μm, painting radius R = 50 mm, painting distance h = 100 mm, numbers of triangular facets N = 1566, the length of each segment $dk$ = 50 mm, number of the subdivided segments m = 10, and weight vector $\omega = (0.5, 0.5)^T$. When the gun paints at uniform speed, v = 256 mm/s (the optimization speed on the plane). Take v = 256 mm/s as the initial value of the algorithm iteration when performing optimized spray painting. The following optimization experiments are carried out in the U-direction path and the V-direction path, respectively. The process of spray painting experiment in the laboratory is shown in Figure 13. After the spray painting experiment, the paint thickness is measured by a paint thickness gauge. The paint thickness curve of the 432 sampling points along the U-direction path is shown in

Figure 14 and the paint thickness curve of the 402 sampling points along the V-direction path is shown in Figure 15. The experimental results are shown in Table 2.

After the analysis of the experimental results, we can learn that spray painting along the U-direction and the V-direction can both meet the spray painting requirement after the optimization of spray painting trajectory. That is, the error of paint thickness is within the allowable range. However, it can be seen that the spray painting effect along the U-direction path is better and the spray painting efficiency is higher for the workpiece. It also can be seen that the shape of the workpiece surface should be fully considered in the planning of painting path, and the direction of the painting path may have a certain impact on the spray painting effect and efficiency.

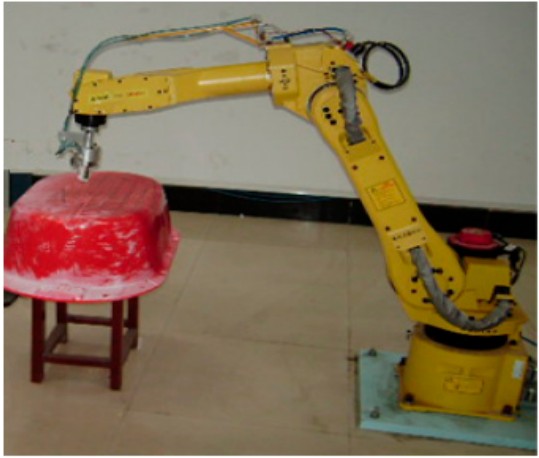

**Figure 13.** Spray painting Experiment in the Laboratory.

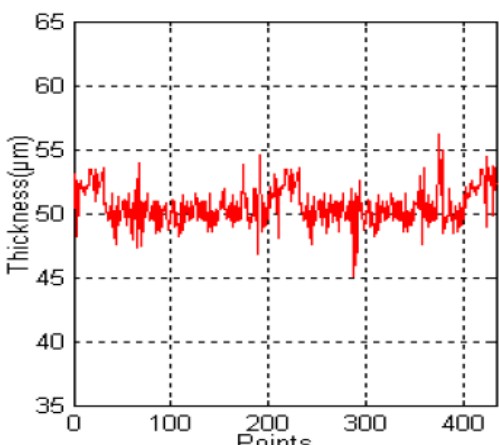

**Figure 14.** The Paint thickness Curve Spray painting along the U-direction Path.

**Table 2.** The Experimental Results of Spraying.

|  | Optimized Spray Painting along U-Direction Path | Optimized Spray Painting along V-Direction Path |
| --- | --- | --- |
| Average (μm) | 51.2 | 52.1 |
| Maximum (μm) | 56.2 | 58.3 |
| Minimum (μm) | 45.1 | 43.0 |
| Painting time (s) | 83 | 95 |

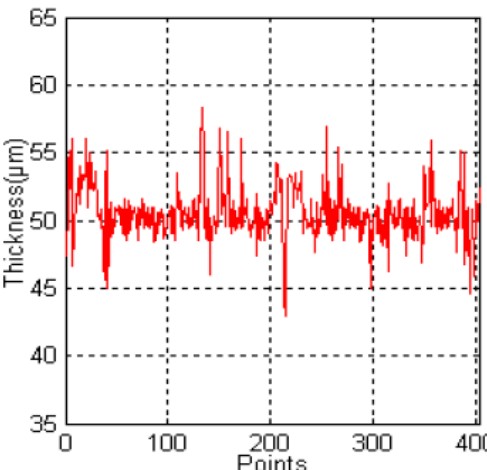

**Figure 15.** The Paint thickness Curve Spray painting along the V-direction Path.

*6.2. Spray Painting Experiment*

Taking a certain brand of automobile body as the spraying target, the feasibility of automatic spraying trajectory optimization on the Bézier surface was simulated. As shown in Figure 16, taking the automotive body of a brand as the paint objective and taking the U direction as the spraying direction. Four ABB robots are using for painting at the same time. After the painting is completed, the coating thickness of the sample points on the surface of the car body is measured by the paint drying and the professional coating thickness tester.

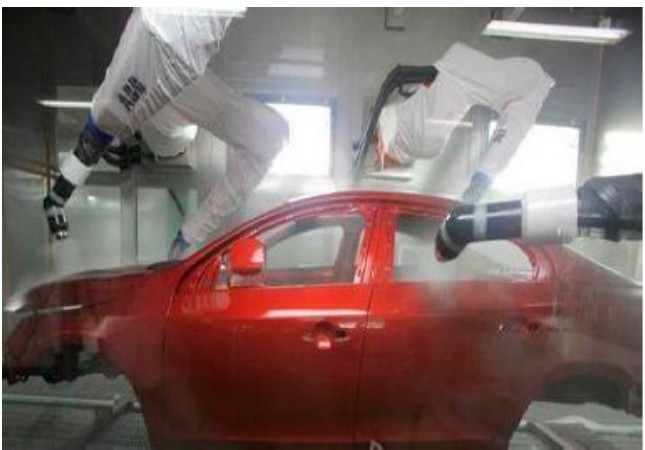

**Figure 16.** The robotic spray painting experiment.

In the spray experiment, the ideal paint thickness is $qd$ = 50 µm, maximum allowable error $qw$ = 10 µm, painting radius R = 50 mm, painting distance h = 100 mm, and painting speed V = 256 mm/s (the optimization speed on the plane) when performing uniform spray painting. We take 400 discrete points evenly on the workpiece surface after the spray painting operation. The paint thickness curve is shown in Figure 17 after using a paint thickness gauge to measure the paint thickness at the discrete points.

Based on the analysis of the experimental results, it can be seen that the average spray thickness is 51.1 µm, the thickness of the maximum coating is 58.1 µm, the minimum coating thickness is 44.2 µm, and the spray time spent by the robot is about 99 s, which is better than the general spraying robot. After the spraying trajectory is normalized based on the Bézier surface, the spraying requirements can be met along the path spraying, that is, the coating thickness deviation is within the allowable range.

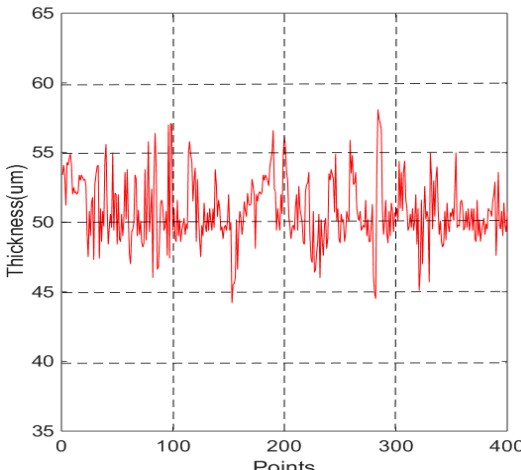

**Figure 17.** Material Thickness of Random Chosen Points for U-direction Trajectory.

## 7. Conclusions

In this paper, an automatic spray trajectory optimization method on Bezier surface is proposed. Firstly, the Bézier surface is analyzed and the method for searching the optimal initial trajectory of the spray painting robot is given according to the features of the Bézier triangular surface. Subsequently, the spray painting model of Bézier surface is established, and the mathematic expression of paint thickness at a certain point on Bézier surface is given. Finally, the optimized trajectory of Bézier surface is obtained by using the ideal point method in the mathematical programming with the uniformity of paint thickness and the shortest spray painting time as optimization objectives along the specified spray painting path. The biggest advantage of this method is that a good initial path of automatic spraying is determined at the beginning of the spraying process, which can significantly provide uniformity of coating thickness, that is, improve the spraying effect. Finally, the effectiveness and practicability of the proposed method are verified by an example verification and spraying experiment.

**Author Contributions:** W.C. and Y.T. conceived and designed the experiments; W.C. performed the experiments; H.G. analyzed the data; J.L. contributed reagents/materials/analysis tools; J.L. wrote the paper.

**Funding:** This research is supported by the Postgraduate Research & Practice Innovation Program of Jiangsu Province (KYCX18_2332), National Natural Science Foundation of China (51505193, 61503162), Project funded by China Postdoctoral Science Foundation (2016M601691), Industrial Foresight Project of Zhenjiang City (GY2018018) and The six highest peak of talent in Jiangsu Province (2016-GDZB-021).

**Conflicts of Interest:** The authors declare no conflict of interest.

## Annotation

| | |
|---|---|
| $T_1, T_2, T_3$ | Clockwise vertex of any given triangle. |
| $P$ | Any point in the plane where the triangle $T_1 T_2 T_3$ is located. |
| $[T_1 T_2 T_3]$ | Represents the directed area of the triangle $T_1 T_2 T_3$. |
| $T$ | The interior of the triangle $T_1 T_2 T_3$ with the boundary. |
| $S_n(T)$ | Make the $n$-time subdivision of the coordinate triangle $T$. |
| $B_{i,m}(u), B_{j,n}(v)$ | The u-direction m-time and v-direction n-time Bernstein basis function. |
| $B(0,0), B(1,0), B(0,1), B(1,1)$ | The four corner points of the Bézier grid. |
| $V_{0,0}, V_{m,0}, V_{0,n}, V_{m,n}$ | The four corner points of the Bézier surface. |
| $\alpha_0$ | The smooth initial trajectory. |
| $C_{st}$ | A segment on the smooth initial trajectory. |
| $\alpha_0(t_0), \alpha_0(t_1)$ | The two endpoints of $C_{st}$. |
| $\gamma_{t_0}, \gamma_{t_1}$ | The geodesic lines. |
| $C_{st}$ | The offset curve of line segment. |
| $\Delta$ | The offset distance. |
| $C_{of}$ | The offset curve with a distance $\Delta$ of $C_{st}$ |

| | |
|---|---|
| $\phi$ | A region enclosed by boundaries $C_{st}$, $C_{of}$, $\gamma_{t_0}$ and $\gamma_{t_1}$ |
| $C_{\text{dia}}$ | Any smooth curve connecting $\gamma_{t_0}(\Delta)$ and $\gamma_{t_1}(0)$ |
| $\phi_1$ | The region bounded by $C_{st}$, $\gamma_{t_0}$ and $C_{\text{dia}}$ |
| $\phi_2$ | The region bounded by $C_{of}$, $\gamma_{t_1}$ and $C_{\text{dia}}$ |
| $\partial\phi_1$ | The boundary of $\phi_1$ |
| $\partial\phi_2$ | The boundary of $\phi_2$ |

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
