# Peer review of "Automatic Spray Trajectory Optimization on Bézier Surface"

_electronics, doi:10.3390/electronics8020168_

Round 1
Reviewer 1 Report
1- The authors published 2 papers on similar data with several other conference papers, one that is very close to this study is entitled "Trajectory Optimization of Spray Painting Robot for Complex Curved Surface Based on Exponential Mean Bézier Method, Mathematical Problems in Engineering, Vol. 2017" could the authors go over the differences in the study/design between the 2 publications and identify why there is a need to have a 4th paper with little changes in the results.
2- please compare the results between the two pervious publications "Coatings, Sep. 2017 and Mathematical Problems in Engineering, Nov., 2017"
Reviewer 2 Report
- Given the number of variables in the equation, a nomenclature page defining all the variables might be required
- Fig 3. needs improvement, its hard to read the variables
Round 2
Reviewer 1 Report
The authors rebuttal to my comments need to be incorporated into the paper. it will make it stronger.
The revisions made in the current version made it much stronger than the original manuscript. I still have reservations on comparing the authors previous work with the current work that need to illustrated to show the improvement made by the authors from one method to another.
Author Response
Dear Editors and Reviewer:
Thank you for your letter and for the reviewers’ comments concerning our manuscript entitled “Automatic spray trajectory optimization on Bézier surface” (ID: 404966). Those comments are all valuable and very helpful for revising and improving our paper, as well as the important guiding significance to our researches. We have studied these comments carefully and have made corrections which we hope meet with approval. The main corrections in the paper and the responds to the reviewer’s comments are as flowing:
Responds to the reviewer’s comments:
Point 1:The authors rebuttal to my comments need to be incorporated into the paper. it will make it stronger. The revisions made in the current version made it much stronger than the original manuscript. I still have reservations on comparing the authors previous work with the current work that need to illustrated to show the improvement made by the authors from one method to another.
Response 1:Thank you for your valuable advice. According to your helpful advice, We have added a supplement to the article. In addition, the difference between the work of this article and the previous work is explained in detail.